# Synthesis, Crystal Structure and Magnetic Properties of a Trinuclear Copper(II) Complex Based on P-Cresol-Substituted Bis(α-Nitronyl Nitroxide) Biradical

**DOI:** 10.3390/molecules27103218

**Published:** 2022-05-18

**Authors:** Sabrina Grenda, Maxime Beau, Dominique Luneau

**Affiliations:** Laboratoire des Multimatériaux et Interfaces (UMR 5615), Université Claude Bernard Lyon 1, 69100 Villeurbanne, France; beau-maxime@outlook.fr

**Keywords:** nitronyl nitroxide biradical, molecular magnetism, copper complex

## Abstract

Trinuclear copper(II) complex [Cu^II^_3_(NIT_2_PhO)_2_Cl_4_] was synthesized with p-cresol-substituted bis(α-nitronyl nitroxide) biradical: 4-methyl-2,6-bis(1-oxyl-3-oxido-4,4,5,5-tetramethyl-2-imidazolin-2-yl)phenol (NIT_2_PhOH). The crystal structure of this heterospin complex was determined using single-crystal X-ray diffraction analysis and exhibits four unusual seven-membered metallocycles formed from the coordination of oxygen atoms of the N-O groups and of bridging phenoxo (µ-PhO^−^) moieties with copper(II) ions. The crystal structure analysis reveals an incipient agostic interaction between a square planar copper center and a hydrogen-carbon bond from one methyl group carried on the coordinated nitronyl-nitroxide radical. The intramolecular Cu∙∙∙H-C interaction involves a six-membered metallocycle and may stabilize the copper center in square planar coordination mode. From the magnetic susceptibility measurements, the complex, which totals seven S = 1/2 spin carriers, has almost a ground state spin S = 1/2 at room temperature ascribed to strong antiferromagnetic interaction between the nitronyl nitroxide moieties and the copper(II) centers and in between the copper(II) centers through the bridging phenoxo oxygen atom.

## 1. Introduction

The coordination chemistry of nitronyl nitroxide free radicals has played a major role in the development of molecule-based magnetic materials following the so-called metal-radical approach pioneered thirty years ago [1]. Several well-known and good reasons make nitronyl nitroxide free radicals attractive organic spin carriers for magnetic coordination compounds. One is the quite easy synthesis of the nitronyl nitroxide (NN) moiety, which can be incorporated into many chemical groups in the α-position. This is usually done by the Ullman procedure via the reaction in methanol of 2,3-bis(hydroxylamino)-2,3-dimethylbutane with an appropriate aldehyde [2]. In addition, a palladium-catalyzed cross-coupling reaction of (nitronyl nitroxide-2-ido)(triphenylphosphine)gold(I) complex [Ph_3_P-Au-NN] with aryl halides has also been developed to directly graft the NN moiety onto aromatic rings [3]. Therefore, an almost unlimited number of open-shell molecules may be synthesized. Thus, nitronyl nitroxide have been grafted on alcanes [4], pyridine [5,6,7,8,9,10], imidazole [11], triazole [12], bipyridine [13], phenol [14,15,16,17], pyrene [18], azulene [19], phthalocyanine [20], porphyrin [21], cyclotriphosphazene [22], or even on graphene [23]. Most importantly, nitronyl nitroxide has one of the best stabilities among the free radicals, and this is retained in most of its derivative so that they can generally be handled under mild but normal conditions, even when coordinated with most metal ions. Although the NO group is a weak Lewis base, coordination is effective by using electron-withdrawing ancillary ligands on the metal center, such as hexafluoroacetylacetone, or by incorporating the NO group into a chelate so that such extra ligand can be removed from the coordination sphere [11]. The nitronyl nitroxide stability in coordination compounds could be tempered after the discovery of valence tautomerism in some 2D coordination polymers with manganese(II), where nitronyl nitroxide radical is reduced on cooling [24,25,26]. However, the process reverses on reheating. To end, the two NO groups of the nitronyl nitroxide (NN) moiety on which the unpaired electron is equally delocalized [27] make it a bridging-ligands but open-shell molecule. This makes it easy to build heterospin systems in which direct magnetic interactions operate between the metal and radical spin carriers. This has provided many magnetic systems [28] among which molecular magnets [29,30,31], single-molecule magnets [31,32,33,34,35,36], or single-chain magnets [37,38,39,40,41]. However, that is not all. Research on metal-nitronyl nitroxide systems has led to the discovery of two other phenomena out of the classical magnetism relying on the interplay of magnetic interaction. These are molecular-spin transitions in some copper(II) complexes [42,43,44] and valence tautomerism in some manganese(II) coordination polymers [24,25,26]. Metal-nitronyl nitroxide coordination chemistry is, therefore, a versatile tool for different types of bistable systems [45].

As for most coordination chemistry, such a development would not have been possible without the crystallography tool, mainly single crystal X-ray diffraction. Crystal structure has indeed been essential to watch the molecular arrangement and to further understand the magnetic properties through the analysis of the structural features, as illustrated in the present paper.

Among metal-nitronyl nitroxide coordination compounds, those with copper(II) are singular regarding the magneto-structural relationships. Indeed, because of the lonly copper(II) magnetic orbital, the copper-radical magnetic interaction is very sensitive and dependent on the coordination mode. It can be either antiferromagnetic or ferromagnetic, depending on whether the coordination mode favors or not the overlap of the magnetic orbitals of the copper(II) with those of the radical [1]. This may even be temperature dependent and give rise to molecular spin transition. [42,43,44].

This versatile but richness of magnetic behaviors prompt us to investigate complexation of copper(II) by bis-nitronyl nitroxide diradical 4-methyl-2,6-bis(1-oxyl-3-oxido-4,4,5,5-tetramethyl-2-imidazolin-2-yl)phenol [14] (NIT_2_PhOH, Figure 1). Herein, we report the reaction with copper(II) chloride to afford a heterospin neutral complex [Cu^II^_3_(NIT_2_PhO)_2_Cl_4_] where this diradical (NIT_2_PhOH) acts both as a bridging and chelating ligand. The complex has been isolated in the crystalline solid state, and the temperature dependence of the magnetic susceptibility has been analyzed based on the structure characterized by single-crystal X-ray diffraction.

## 2. Results

### 2.1. Synthesis of the Inorganic Complex

Trinuclear copper complex [Cu^II^_3_(NIT_2_PhO)_2_Cl_4_] (NIT_2_PhOH = 4-methyl-2,6-bis(1-oxyl-3-oxido-4,4,5,5-tetramethyl-2-imidazolin-2-yl)phenol) was synthesized as brown squared crystals from a mixture in methanol at room temperature of copper(II) chloride and NIT_2_PhOH in a 2:1 molar ratio.

### 2.2. Crystal Structure

[Cu_3_(NIT_2_PhO)_2_Cl_4_] crystallizes in the monoclinic P2_1_/n space group. Crystallographic data are reported in Table 1. The asymmetric unit exemplified in Figure 1 comprises all the compounds. It is made of three crystallographically independent copper(II) ions coordinated with two deprotonated bis-nitronyl-nitroxide ligands, so-called hereafter diradicals A and B involving their phenoxide oxygen atoms and the nitronyl nitroxide moieties (NN). The copper ions have their coordination sphere completed by four chloride ions among which two are bridging. This makes the complex electrically neutral. A simplified representation of the coordination mode of the two diradicals is also shown in Figure 2.

In diradical A and B, as generally observed, the phenoxide oxygen atoms are bridging for Cu2 and Cu1 (Diradical A) and for Cu1 and Cu3 (Diradical B). In both cases, nitronyl nitroxide (NN) moieties complete the coordination sphere of the copper(II) centers but depending on Diradical A or Diradical B, these NN moieties expose two types of coordination modes. In Diratical A, one of the NN moiety is chelating for Cu2, while the second NN moiety is chelating for Cu1, so that Diradical A may be viewed as bridging the Cu2 and Cu1 through the phenoxide oxygen atoms and through the NN moieties. In Diradical B, both NN moieties play differently and are solely chelated to Cu3. In all cases, the NN ligand displays the unusual seven-membered metallocycle reported elsewhere in complexes of closely related phenol substituted nitronyl nitroxide [15,16,17,46,47].

Cu1 and Cu3 have a coordinance of five, but Cu2 has an uncommon coordinance of four. Bond lengths and angles constitutive of the environment of the three copper(II) are listed in Table 2.

**Cu1** (Figure 2, Table 2) assumes an intermediary environment in between square planar pyramidal and trigonal bipyramidal with an Addison parameter of τ_5_ = 0.46 [48]. The shortest bond length range 1.934–1.947 Å is found for Cu1-O bonds with oxygen atoms of one nitronyl nitroxyde radical of diradical A and the phenoxide oxygen atoms of bisradical A and B. The longest bond lengths 2.2596–2.5870 Å are found for Cu1-Cl bonds. As the length of the Cu1-Cl3 biding (2.59 Å) is the longest, we assume that the geometric environment might be a distorted square planar pyramidal with O5B, Cl2, O5A, O1A forming the base and Cl3 occupying the apical position, in agreement with an elongation Jahn Teller distortion as expected.

The length of the apical Cu1-Cl3 bonding (2.59 Å) is shorter than typical Cu-Cl bonds lengths found in the literature. For apical bridging, chloride values between 2.709 Å and 2.785 Å are found [49,50]. The equatorial Cu1-Cl2 (2.26 Å) bond length is in the range of typical Cu-Cl bonds found in the literature with values between 2.264 Å and 2.300 Å for a bridging chloride ion [50,51]. For Cu1-O5A (1.93 Å) and Cu1-O5B (1.94 Å), bond lengths are consistent with a µ-hydroxo character of the bridge with values between 1.923 Å and 1.97 Å, as reported in the literature [52,53]. Finally, the equatorial Cu1-O1A (1.93Å) distance is in the range for Cu-O_NO_ bonding where values between 1.95 Å and 2.06 Å are reported [47].

**Cu2** (Figure 2 and Figure 3, Table 2) assumes a highly distorted square planar geometry with a Houser–Okuniewski parameter τ_4_ = 0.28 [54,55,56] and seems to be stabilized in its axial position by an intramolecular interaction with a hydrogen (H18A) from the methyl group (C18) carried on the NN-Radical moieties. This intramolecular interaction, together with the oxygen atom of nitroxide group (O3A), makes a chelate. This affords a six-membered metallacycle and brings the methyl carbon (C18) close to Cu2, as shown in Figure 3. These structural features suggest an incipient Cu(II) ∙∙∙ H-C agostic interaction [57,58]. The Cu2∙∙∙H18A and Cu2∙∙∙C18 interatomic distances are respectively 2.6656(7) Å and 3.1112(6) Å with an angle M∙∙∙H18A-C18 of 125.076(3)°, which fall in the range of structural critters for an agostic interactions parameters[59,60].

**Cu3** (Figure 4, Table 2) assumes a distorted trigonal bipyramidal geometry with an Addison parameter of τ_5_ = 0.84 [48]. The shortest bond length range 1.909–2.110 Å is found for Cu3-O bonds with oxygen atoms of the two nitronyl nitroxyde radical of diradical B and its phenoxido oxygen atoms. The longest bond lengths 2.1585–2.6439 Å are found for Cu1-Cl bonds. As the bond length of axial Cu3-Cl1 biding (2.16 Å) and axial Cu3-O5B biding (1.909 Å) are shorter than equatorial Cu3-Cl2 (2.64 Å), Cu3-O1B (2.110 Å), and Cu3-O3B (1.966 Å), the environment of copper Cu3 can be described by an unusual compression Jahn–Teller distortion. Indeed, the equatorial Cu3-Cl2 (2.64 Å), Cu3-O1B (2.110 Å), and Cu3-O3B (1.966 Å) are much longer than the typical bond length, as described before, and axial Cu3-Cl1 biding (2.16 Å) is highly shorter than typical Cu-Cl length for a terminal chloride atom where a value of 2.267 Å is reported [50].

Bond lengths of the N−O group within the four NN moieties are comprised in the range 1.254(6)–1.300(5) Å (Table 2) with small elongation (0.03 Å) of the coordinated ones and are characteristic of free and coordinated nitroxide radicals [4,5,6,7,8,9,10,11,12,13,14,15,16,17,18,19,20,21,22,23]. The O-N-C-N-O moieties are strictly planar (Table 3), as expected due to electron delocalization. These structural features demonstrate the persistence of the four nitronyl nitroxide radicals in [Cu_3_Cl_4_(NIT_2_PhO)_2_]. The dihedral angles between the O-N-C-N-O least-square plane and with attached phenyl ring (φ) are quite close for the four NN moieties in contrast with the free radical for which 38°, 64°, and 72° were found for φ [14].

### 2.3. Magnetic Behaviour

The temperature dependence of the product of the magnetic susceptibility with the temperature (χ_M_T) of [Cu^II^_3_(NIT_2_PhO)_2_Cl_4_] is shown in Figure 5. At 370 K, χ_M_T is 0.57 emu·K·mol^−1^, then, upon cooling, it decreases almost continuously down to 0.40 emu·K·mol^−1^ at 75 K. From there, it remains almost constant down to 30 K and then, it decreases abruptly to reach 0.14 emu·K·mol^−1^ at 3 K. The high temperature χ_M_T value is very much lower than the expected value (~2.8 emu·K·mol^−1^) for the seven magnetically independent spins S = 1/2, taking into account the three copper(II) ions and the four nitronyl nitroxide radicals. This, together with the continuous decreasing upon cooling down to a plateau with a χ_M_T value corresponding to one resultant spin S = 1/2, is indicative of strong antiferromagnetic interactions operating in this hetero-spin complex.

The understanding of the magnetic behavior at the molecule level requires considering a complex and multiple exchange interactions network, as schematized in Figure 6: *(i) Interaction in between the nitronyl nitroxide (NN) moieties through the phenyl ring (J_NN-NN_).* From the study of the pure diradical (NIT_2_PhOH), it was estimated to be weak and ferromagnetic (2J/k = 12 K) [14]. *(ii) Interaction between the NN moieties and next nearer Cu(II) through the phenyl rings via the bridging phenoxo oxygen atoms (J’_NN-Cu_).* Herein, this exchange interaction may be significant but weaker than the exchange interaction between copper centers through the bridging phenoxide oxygen atom and chloride atoms and also compared with direct interactions between copper centers and N-oxide moieties. [16,17,58]. *(iii) Cu(II) ∙∙∙Cu(II) super-exchange interaction through the phenoxide oxygen atoms and chloro bridges (J_Cu-Cu_).* According to the study of the crystal structure, we can determine the nature of the magnetic orbitals of metal centers. For Cu1 and copper Cu2, it was assumed to be respectively an elongated square planar pyramid and a square plane. In these two cases, the $dx^{2}-y^{2}$ orbitals define the magnetic orbitals. For Cu3, the environment is a compressed trigonal bipyramid, which defines $dz^{2}$ orbital as the magnetic orbital. Regarding these results, the magnetic exchange is essentially governed by the $3dx^{2}-y^{2}$(copper)–2p(oxygen) and the $3dz^{2}$(copper)–2p(oxygen) antibonding overlaps. Considering the Cu-Cl-Cu bond angles and Cu-Cl bond lengths, we do not expect the chloro bridges to mediate significant interaction (Table 2 and Table 3). Magneto-structural correlations of µ-hydroxo-bridged copper(II) binuclear complexes have shown that the exchange interaction depends on Cu-Cu distance, Cu-O-Cu angle [59,60,61]. Here, the copper centers are separated by 3.212 Å for Cu1-Cu2 and 3.286 Å for Cu1-Cu3 and considering the quite large Cu-O-Cu angle of the phenoxido bridge (Table 2), this should give strong antiferromagnetic interaction [62]. The exchange interaction parameter (*J*) may be estimated to be greater than 700 cm^−1^ from the Hatfield’s correlation 2*J* = 74.53Φ−7270 cm^−1^ and 2*J* = −4508d + 13018, taking the Cu-O-Cu angle (Φ) or Cu-Cu distance (d), respectively [63,64,65]. These interactions may be lower here because of the coordinated chloride ions. Indeed, from a previous study of unsymmetrical µ-hydroxo copper(II) complexes it has been reported that the nature of the exogenous bound ligands with varying electronegativity influences the exchange interactions within copper centers. It was found that bonding chloride ions greatly lower the antiferromagnetic couplings because they decrease the electron density on the coppers [66]. *(iv) Direct interaction between the NN moieties and Cu(II) spin carriers (J_NN-Cu_).* According to the crystal structure, for the three copper centers, nitroxide radicals are equatorially coordinated to the square plane (Cu2), square planar pyramid (Cu1), and trigonal bipyramid (Cu3). For Cu1 and Cu2, this is to favor antiferromagnetic interaction [1]. Moreover, the Cu-O-N angles (126.4°–112.7°) and the small dihedral (δ) angle between the Cu-O-N and O-N-C-N-O least-square planes (Table 3) cause substantial overlap between the π* orbital of the nitroxide radical and the 3$dx^{2}-y^{2}$ orbitals of copper Cu1 and Cu2, leading to large antiferromagnetic couplings (−500 cm^−1^) [1,62,66,67,68]. For Cu3, coordination of the nitroxide radical in the equatorial plane is not expected to cause any overlap with the *dz^2^* magnetic orbital of the trigonal bipyramid. This rules out any significant magnetic interaction.

With this scheme of possible interactions in mind, we tried to fit the temperature dependence of the product of the magnetic susceptibility with temperature (χ_M_T) using the PHI program [69]. All tentatives taking into account seven spin S = 1/2, using a different set of above interactions, were unsuccessful. The best fit was obtained only when considering a three spin system and three coupling constants: *J_12_*= −265(6) cm^−1^; *J_13_* = −208(6) cm^−1^; *J_23_* = −30(1) cm^−1^; g1 = 2.32(1); g2 = g3 = 2; and z*J’* = −3.07(8) cm^−1^. This is in agreement with the low χ_Μ_Τ values reaching high temperature (370k), and it means that already in this temperature region, there are only three effective spins S = 1/2. One could think the nitronyl nitroxide radicals have been killed, and they are the three copper(II) ions only. This has to be ruled out because, as we have seen, the structural features demonstrate without any doubt the persistence of the four nitronyl nitroxide moieties. Moreover, the complex is neutral and this also rules out that the radicals could have been reduced or oxidized. Therefore, the unique possibility is that four spins S = 1/2 became silent because of antiferromagnetic couplings larger than the thermal energy. In agreement with the previous discussion of possible interactions, this is attributed to the antiferromagnetic coupling of each of the copper Cu1 and Cu2 with their attached nitronyl nitroxide from biradical A. As discussed above, the structural features are expected to result in large antiferromagnetic interaction (~500 cm^−1^). This makes the four spins S = 1/2 comprising Cu1 and Cu2 and Diradical A silent. We may think this primes on the Cu—Cu interactions (*J_Cu-Cu:_*
Figure 6), which may be lower due to electron withdrawing effect of coordinated chloride ions, as discussed above. The resulting three spin S = 1/2 system is thus attributed to Cu3 interacting with the two nitroxide of biradical B. As we discuss above, Cu3 is in bipyramid trigonal coordination geometry, and this is not expected to cause direct antiferromagnetic interaction with the *dz^2^* magnetic orbital. In that case, the interaction of the two nitronyl nitroxide moieties of Diradical B with Cu3 is indirect and proceeds via the phenoxido oxygen atom (*J’_NN-Cu_*: Figure 6). This is ascribed to *J_12_* and *J_13_*, with g1 holding for Cu3. Then, *J_23_* is ascribed to the interaction between the nitronyl nitroxide radical via the phenyl ring, as this is not expected to be large (*J_NN-NN_*: Figure 6). The interaction (*J_23_*) is moderate but antiferromagnetic in contrast with the ferromagnetic one found for the free radical [14]. This is attributed to the difference in the dihedral angle between the O-N-C-N-O and phenyl least-square planes combined with the change in spin density distribution consecutive to coordination.

## 3. Materials and Methods

### 3.1. Materials

All chemicals and solvents were purchased as analytical grade and were used without further purification. 4-methyl-2,6-bis(1-oxyl-3-oxido-4,4,5,5-tetramethyl-2-imidazolin-2-yl)phenol (NIT_2_PhOH) was synthesized following a reported procedure [14].

### 3.2. Synthesis of Complex Cu_3_(NIT_2_Ph)_2_Cl_4_

10 mL of a methanol solution of CuCl_2_ (64 mg, 0.48 mmol), which was previously dried in a desiccator, was added to a 10 mL of a methanol solution of (NIT_2_PhOH) (100 mg, 0.24 mmol). The dark brown solution was left for crystallization by slow evaporation. Dark brown crystals, which appeared after three weeks, were isolated by filtration and then washed with ethanol. The complex is insoluble in most usual solvents, which preclude such measurements as UV-vis and molar conductivity, as reported elsewhere [66]. Yield: 43.9 mg (0.04mmol, 33% in term of NIT_2_PhOH ligand). Elemental analysis (%): C, 42.86; H, 4.98; Cu, 16.29; N, 9.65; Calculated for C_42_H_58_Cl_4_Cu_3_N_8_O_10_ (%): C, 43.21; H, 5.01; Cl, 12.15; Cu, 16.33; N, 9.60; O, 13.70; IR spectrum (υ/cm^−1^) at 293(2) K: 2987 w, 2938 w, 1431 m, 1338 m, 1312 s, 1214 s, 1171 s, 1145 m, 1057 m, 940 m, 870 m, 798 m, 737 m, 598 m, 547 m, 444 s, 424 s.

### 3.3. Single-Crystal X-ray Diffraction

Single-crystal diffraction data were collected on a Xcalibur Gemini diffractometer with graphite monochromated Mo Kα radiation (λ = 0.71073 Å), using the related analysis software [70]. Absorption correction has not been performed because it caused a significant decrease in data quality: Increase of Rint value and decrease of the rate of completeness. The structures were solved using the SHELXT program [71] and refined by full-matrix least-square methods on F² with the 2018 version of SHELXL program [72] on OLEX2 software [61,73,74]. All non-hydrogen atoms were refined with anisotropic displacement parameters. Hydrogen atoms belonging to carbon atoms were placed geometrically in their idealized positions and refined using a riding model. Crystallographic data are presented in Table 1. Selected bond lengths and bond angles are collected in Table 2 and Table 3. Crystallographic data for the structures have been deposited with the Cambridge Crystallographic Data Centre as supplementary publication nos: CCDC 2132162. Copies of the data can be obtained free of charge on application to CCDC, 12 Union Road, Cambridge CB2 1EZ, UK (fax, +44-(0)1223-336033; or e-mail, deposit@ccdc.cam.ac.uk).

### 3.4. Magnetic Measurements

Magnetic susceptibility data (2–300 K) were collected on powder samples using a SQUID magnetometer (Quantum Design model MPMS-XL) in a 1T applied magnetic field. A magnetization isotherm (2 K) was measured between 0–5 T. All data were corrected for the contribution of the sample holder and diamagnetism of the samples estimated from Pascal’s constants [61,75,76].

## 4. Conclusions

This paper reports the synthesis of a neutral trinuclear copper(II) complex [Cu^II^_3_(NIT_2_PhO)_2_Cl_4_] affords by coordination with biradical, 4-methyl-2,6-bis(1-oxyl-3-oxido-4,4,5,5-tetramethyl-2-imidazolin-2-yl)phenol (NIT_2_PhOH) completed by cloride. The crystal structure of this heterospin complex reveals a complicated arrangement in which two copper(II) ions, Cu1 and Cu2, are coordinated to each of the two nitronyl nitroxide moieties on one diradical (A). The third copper(II) ion is coordinated to both nitronyl nitroxide moieties of a second diradical (B). The three coppers are also bridged by the phenoxido oxygen atoms of deprotonated diradicals together with some of the cloride ions. The crystal structure analysis reveals an incipient agostic interaction between a square planar copper center and a hydrogen atom from one methyl group carried on the coordinated nitronyl-nitroxide radical. From the magnetic susceptibility measurements, this seven S = 1/2 spin carrier complex behaves as a three spin resulting systems S = 1/2 system. This is ascribed to strong antiferromagnetic direct interaction between the nitronyl nitroxide moieties of diradical A and their coordinated copper(II) ion, Cu1 and Cu2. This makes these four spin S = 1/2 silent, even at high temperature (370K), evidencing only the coupling within Cu3 and diradical B. This is a novel example of how much coordination nitronyl nitroxyde can generate strong magnetic interaction.

## Data Availability

Data are available from the authors D.L. and S.G.

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
