# Peer review of "Synthesis, Crystal Structure and Magnetic Properties of a Trinuclear Copper(II) Complex Based on P-Cresol-Substituted Bis(α-Nitronyl Nitroxide) Biradical"

_molecules, 2022, doi:10.3390/molecules27103218_

Round 1

Reviewer 1 Report

The manuscript reports on the synthesis and crystallographic characterization of trinuclear copper(II) complex [CuII3(NIT2PhO)2Cl4], as well as its magnetic properties. The manuscript can be accepted for publication in Molecules, after consideration of the following issues:

1) IR and UV-Vis spectroscopy should be used for characterization of the complex. Spectroscopic data should be discussed and compared with those for the structurally similar complexes.

2) Molar conductivity of the complex should be determined.

3) Elemental analysis of the complex should be performed, and the data should be given in the Materials and Methods section.

4) The oxidation state of the copper should be written through the manuscript. Moreover, the copper and its oxidation state should be written without space.

5) Table  3 is firstly mentioned un the text. This should be corrected.

6) The rationale behind the choice of the metal ion should be given in the Introduction.

7) The French words (Table 3) should be translated in English.  

Author Response

The manuscript reports on the synthesis and crystallographic characterization of trinuclear copper(II) complex [CuII3(NIT2PhO)2Cl4], as well as its magnetic properties. The manuscript can be accepted for publication in Molecules, after consideration of the following issues:

  • IR and UV-Vis spectroscopy should be used for characterization of the complex.

Author’s responses: IR spectrum data were added in the Materials and Methods section. This was not possible for UV-vis spectra due to bad quality of spectrum in the solid state.

  • Spectroscopic data should be discussed and compared with those for the structurally similar complexes.

Author’s response: We found no spectroscopic data of closely related complexes except some copper complexes with Schiff bases derivate from 2,6-diformyl-4-methylphenol (Transition Metal Chemistry, 2003, 28, 447-454 ) but not so close due to the nitronyl nitroxide moities.

  • Molar conductivity of the complex should be determined.

Author’s response: the complexes is insoluble in most solvent which preclude measurements of the conductivity. This now stated in the Materials and Methods section.

  • Elemental analysis of the complex should be performed, and the data should be given in the Materials and Methods section.

Author’s response: Elemental analysis of the complex were performed, and the data are now given in the Materials and Methods section.

  • The oxidation state of the copper should be written through the manuscript. Moreover, the copper and its oxidation state should be written without space

Author’s response: The oxidation state of the copper is now written through the manuscript and we check carefully that no space come in between copper and its oxidation state.

  • Table  3 is firstly mentioned un the text. This should be corrected.

Author’s response: Table 3 has been renamed Table 1 and now comes first.

  • The rationale behind the choice of the metal ion should be given in the Introduction.

Author’s response: The reason for choosing copper(II) is now given in introduction.

7) The French words (Table 3) should be translated in English.  

Author’s response: French words was translated in English in Table 3.

Reviewer 2 Report

The work describes the synthesis of a trinuclear copper complex with two biradcal ligands NIT2PhOH. The structure of the compound was studied by X-ray diffraction. All three copper atoms have different coordination environments and different geometries. The NIT2PhOH ligands differ in the way they coordinate Cu. An interesting point is the formation of a short contact Cu2···H18AC18A; the authors attributed it to agostic. The magnetic susceptibility of the powder of the complex has been studied, and it has been shown that the magnetic behavior is best described as a three-spin system. The construction of a scheme of interactions and comparison with structural and literature data allowed the authors to attribute the S=1/2 three-spin system to the Cu3 atom and two nitroxides of the ligand biradical. Due to the strong antiferromagnetic interaction in the Cu1/Cu2/biradical system, four other possible centers remain silent. The work completely describes the structure and magnetic behavior of the new three-core complex and is in the area of interest of the readers of the journal Molecules. The work can be accepted for publication.

There are some small notes:

Page 7. For agostic interaction, the Cu2-C18A distance should also be specified. This type of contacts suggests orientation on the sigma C-H bond.

Page 12. The yield of the compound is given, indicate - is in terms of Cu (taken in excess) or NIT2PhOH ligand?

There are typos in the text, the text should be profred.

In the reference list for many references, the month of publication is given. If necessary, it should be indicated in the language of publication.

References 60 and 67 have an unusual format. It should be checked for correctness.

Author Response

The work describes the synthesis of a trinuclear copper complex with two biradical ligands NIT2PhOH. The structure of the compound was studied by X-ray diffraction. All three copper atoms have different coordination environments and different geometries. The NIT2PhOH ligands differ in the way they coordinate Cu. An interesting point is the formation of a short contact Cu2···H18AC18A; the authors attributed it to agostic. The magnetic susceptibility of the powder of the complex has been studied, and it has been shown that the magnetic behavior is best described as a three-spin system. The construction of a scheme of interactions and comparison with structural and literature data allowed the authors to attribute the S=1/2 three-spin system to the Cu3 atom and two nitroxides of the ligand biradical. Due to the strong antiferromagnetic interaction in the Cu1/Cu2/biradical system, four other possible centers remain silent. The work completely describes the structure and magnetic behavior of the new three-core complex and is in the area of interest of the readers of the journal Molecules. The work can be accepted for publication.

There are some small notes:

Page 7. For agostic interaction, the Cu2-C18A distance should also be specified. This type of contacts suggests orientation on the sigma C-H bond.

Author’s response: We totally agree with this comment. Structural features regarding the agnostic interaction which comprise the Cu2-H18A and Cu2-C18 interactomic distances together with the Cu2-H18A-C18 angle are now specified

Page 12. The yield of the compound is given, indicate - is in terms of Cu (taken in excess) or NIT2PhOH ligand?

Author’s response: the yield is now given, in terms of the NIT2PhOH, in the Materials and Methods section.

There are typos in the text, the text should be profred.

Author’s response: We did our best to remove typos.

In the reference list for many references, the month of publication is given. If necessary, it should be indicated in the language of publication.

Author’s response: Thank you for pointing this. It was a bad configuration of our reference software. We have corrected and removed month of publication.

References 60 and 67 have an unusual format. It should be checked for correctness.

Author’s response: Format of references 60 and 67 have been corrected.

Reviewer 3 Report

The paper is well written, Please consider the following minor points to further improve the MS.

  1. In the abstract the authors mention Cu(II) and then specifying CuII3. I suggest to remove the oxidation state of Cu in the formula. There is no confusion in oxidation state, so it must be avoided in the following text. 
  2. At certain instance the statements need clarifications such as "the complex which totals seven S=1/2 spin carriers....."  authors may check through out the text for clarification. 
  3. Check full stop (.) before or after the [Ref], at least with [1]. Also check [x], [y] or [x], - [y] and revise to [x,y] or [x-y] etc. 
  4.  Check Cu3 in section 3.2, the number 3 might be subscript.  

The work presented in the MS is a careful contribution and this reviewer has no observation to recommend its acceptance for publication in Molecules. 

Author Response

The paper is well written, Please consider the following minor points to further improve the MS.

  1. In the abstract the authors mention Cu(II) and then specifying CuII3. I suggest to remove the oxidation state of Cu in the formula. There is no confusion in oxidation state, so it must be avoided in the following text. 

Author’s response: To fulfill requirement of reviewer 1 the oxidation state of the copper was instead now written through the manuscript.

At certain instance the statements need clarifications such as "the complex which totals seven S=1/2 spin carriers....."  authors may check through out the text for clarification. 

Author’s response: We try our best to clarify our statements. The sentence was revised as followed: “From the magnetic susceptibility measurements, this seven S=1/2 spin carriers complex behaves as a three spin resulting systems S=1/2 system.”

  1. Check full stop (.) before or after the [Ref], at least with [1]. Also check [x], [y] or [x], - [y] and revise to [x,y] or [x-y] etc. 

Author’s response: these have been change accordingly.

  1.  Check Cu3 in section 3.2, the number 3 might be subscript.  

Author’s response: these have been change accordingly.

Round 2

Reviewer 1 Report

The authors improved the manuscript and it can be accepted in the present form.